# Improved Quantum Molecular Dynamics Model and Its Application to Ternary Breakup Reactions

**Junlong Tian [1], Xian Li [2] and Cheng Li [1,\*]**

[1] Guangxi Key Laboratory of Nuclear Physics and Technology, College of Physics and Technology, Guangxi Normal University, Guilin 541004, China

[2] College of Mathematics and Physics, Leshan Normal University, Leshan 614000, China

\* Correspondence: licheng@mail.bnu.edu.cn

**Abstract:** Collisions of very heavy nuclei $^{197}$Au+$^{197}$Au at the energy range of 5–30 $A$ MeV have been studied within the improved quantum molecular dynamics (ImQMD) model. A class of ternary events satisfying a nearly complete balance of mass numbers is selected and we find that the probability of ternary breakup depends on the incident energy and the impact parameter. It is also found that the largest probability of ternary breakup is located at the energy around 24 $A$ MeV for the system $^{197}$Au+$^{197}$Au. The experimental mass distributions and angular distributions for the system $^{197}$Au+$^{197}$Au ternary breakup fragments can be reproduced well by the calculation with the ImQMD model at the energy of 15 $A$ MeV. The modes and mechanisms of ternary and quaternary breakup are studied by time-dependent snapshots of ternary events. The direct prolate, direct oblate, and cascade ternary breakup modes, are manifested and their production probabilities are obtained. The characteristic features in ternary breakup events, three mass-comparable fragments, and the very fast, nearly collinear breakup, account for the two-preformed-neck shape of the composite system. The mean free path of nucleons in the reaction system is studied and the shorter mean free path is responsible for the ternary breakup with three mass comparable fragments, in which the two-body dissipation mechanism plays a dominant role.

**Keywords:** the ImQMD model; ternary breakup; two-preformed-neck

## 1. Introduction

Usually fission proceeds by decay into two fragments of comparable size, while ternary fission means that a third light-charged particle is emitted right at scission from the neck region between the two nascent fission fragments with probabilities at the $10^{-3}$ level. An even much rarer process is quaternary fission where the two main fragments are accompanied by two light-charged particles with probabilities at the $10^{-7}$ level. This kind of ternary or quaternary fission appearing in actinium elements U, Cf, and so on has been studied for decades. For very heavy systems, for instance, Au+Au and U+U, the feature of ternary fission could be very different from "normal" ternary fission. In those systems, there is very clear evidence [1–4] for fission into three mass-comparable fragments and the very fast, nearly collinear breakup processes. In Ref. [1], ternary partitions of a system $^{197}$Au+$^{197}$Au at 15 $A$ MeV were performed in 4 $\pi$ geometry using the multidetector array CHIMERA at LNS Catania. The mass number distributions of fragments were shown according to the mass $A_1$ (the heaviest), $A_2$ (the intermediate), and $A_3$ (the lightest), and the peak of mass number distribution of fragment $A_3$ was around 100. This kind of ternary fission is called "ternary breakup". The features of the ternary breakup reactions explored in these experiments are completely different from the commonly known process of formation of light-charged particles that accompanies binary fission. Ternary breakup is the starting point of multifragmentation in the phenomenon. We have learned that the weak repulsion of the nucleus-nucleus interaction potential after touching configuration,

the strong dissipation, and the strong Coulomb barrier may lead to the existence of the giant composite system for a period of time. From a dynamic point of view, ternary breakup looks like the process of fusion-fission or quasifission. So at a certain energy region, a ternary breakup reaction is inserted between the binary fission and multifragmentation, when the composite system becomes very heavy. It seems to be worthwhile for us to perform a microscopically dynamic study of the mechanism of the ternary breakup with three comparable fragments, which can contribute to deepening our understanding of the dynamics of heavy-ion collisions and testing the theoretical model.

Nuclear fission is known to be a strong-damped process with energy flowing from the collective to the internal single particle excitation energy right up to the point of scission. However, the nature of the nuclear dissipation mechanism is always an important but controversial issue. Especially for these ternary fission could produce three comparable fragments. It is commonly believed that there are two different kinds of energy dissipation mechanisms: one-body [5,6] and two-body dissipation [7,8]. In the one-body process, nucleons collide with the nuclear potential wall generated by a common self-consistent mean field and the two-body dissipation proceeds from collisions between individual nucleons. In the early work [9], Carjan, Sierk, and Nix proposed that observation of the partitioning of the heavy nuclear system might be a suitable way to distinguish these two kinds of dissipation mechanisms. In the case of two-body dissipation, the formation of a large third fragment was predicted in a very heavy nuclear system. On the contrary, in the case of one-body dissipation, the third fragment should be expected to be much smaller. While the peak of mass distribution of fragment $A_3$ is found at about 100 in the ternary breakup experiment work [1]. This result can serve as experimental evidence for clarifying that the two-body dissipation process is more important than that of one-body dissipation in this ternary breakup reaction. But up to now, the microscopic description of these two types of dissipation mechanisms in nuclear dynamics is still not very clear.

In previous papers [10–17], the ternary breakup processes of $^{197}$Au+$^{197}$Au collisions at 15 $A$ MeV have been systematically studied by using the improved quantum molecular dynamics (ImQMD) model, which is a microscopic dynamic model being successfully applied to simulate heavy-ion collisions at low and intermediate energies [18,19]. The calculation results with the ImQMD model can finely reproduce the mass distributions and angular distributions of fragments, and the characteristic features in ternary breakup events explored in the experiments [2,4]. The study shows that those ternary breakup events having the characteristic features found in the experiments happened in the central and semi-central collisions and the composite system has a two-preformed-neck shape, but not at peripheral reactions. The ternary breakup reaction at peripheral reactions belongs to binary breakup with a neck emission.

The paper is reviewed as follows. In Section 2 the ImQMD model is briefly introduced. In Section 3 the mechanism of the ternary and quaternary breakup and the energy dissipation mechanism in Au+Au at 15 $A$ MeV are investigated. Finally, a summary is given in Section 4.

## 2. The ImQMD Model

The improved quantum molecular dynamics (ImQMD) model is successfully used to simulate heavy-ion collisions at low and intermediate energies by making a series of improvements [18,19]. There are three aspects of improvement of the ImQMD model based on the original QMD model [20–23]. First, the surface energy and surface symmetry energy terms are introduced in the potential energy density functional; Second, a system size-dependent wave packet width is introduced; Third, the phase space occupation constraint is adopted [24]. The dissipation, diffusion, and correlation effects are all included in the model. So the ImQMD model is appropriate to study the nuclear reaction mechanism and the energy dissipation mechanism with massive nuclei at low energy.

Let us first briefly introduce the ImQMD model. In the model, the same as in the original QMD model [20–23], each nucleon is represented by a Gaussian wave packet,

$$\phi_i(\vec{r}) = \frac{1}{(2\pi\sigma_r^2)^{3/4}} \exp[-\frac{(\vec{r} - \vec{r}_i)^2}{4\sigma_r^2} + \frac{i\vec{p}_i \cdot \vec{r}}{\hbar}], \tag{1}$$

where $r_i$ and $p_i$ are the centers of the $i$th wave packet in the coordinate and momentum space, respectively. $\sigma_r$ represents the spatial spread of the wave packet. The total N-body wave function is assumed to be the direct product of these coherent states. Through a Wigner transformation, the one-body phase space distribution function for N-distinguishable particles is given by

$$\begin{aligned}
f_i(\vec{r}, \vec{p}) &= \frac{1}{(2\pi\hbar)^3} \int \exp\left(\frac{-i\vec{r}_{12} \cdot \vec{p}_{12}}{\hbar}\right) \phi_i^* \phi_i d\vec{r}_{12} \\
&= \frac{1}{(\pi\hbar)^3} \exp\left(\frac{-(\vec{r} - \vec{r}_i)^2}{2L}\right) \exp\left(\frac{-(\vec{p} - \vec{p}_i)^2 \cdot 2L}{\hbar^2}\right)
\end{aligned} \tag{2}$$

For identical fermions, the effects of the Pauli principle were discussed in a broader context by Feldmeier and Schnack [25]. The approximate treatment of anti-symmetrization is adopted in the ImQMD model by means of the phase space occupation constraint method [24]. The density and momentum distribution functions of a system read

$$\rho(\vec{r}) = \int f(\vec{r}, \vec{p}) d\vec{p} = \frac{1}{(2\pi\sigma_r^2)^{3/2}} \sum_i \exp\left(-\frac{(\vec{r} - \vec{r}_i)^2}{2\sigma_r^2}\right) \tag{3}$$

$$g(\vec{p}) = \int f(\vec{r}, \vec{p}) d\vec{r} = \frac{1}{(2\pi\sigma_p^2)^{3/2}} \sum_i \exp\left(-\frac{(\vec{p} - \vec{p}_i)^2}{2\sigma_p^2}\right) \tag{4}$$

respectively, where the sum runs over all particles in the system. where $\sigma_r$ and $\sigma_p$ are the widths of wave packets in coordinate and momentum space, respectively, and they satisfy the minimum uncertainty relation

$$\sigma_r \cdot \sigma_p = \hbar/2 \tag{5}$$

The propagation of nucleons under the self-consistently generated mean field is governed by Hamiltonian equations of motion:

$$\dot{\vec{r}}_i = \frac{\partial H}{\partial \vec{p}_i} \text{ and } \dot{\vec{p}}_i = -\frac{\partial H}{\partial \vec{r}_i}. \tag{6}$$

The Hamiltonian $H$ consists of the kinetic energy and effective interaction potential energy, i.e.,

$$H = T + U, \tag{7}$$

$$T = \sum_i \frac{P_i^2}{2m} \tag{8}$$

The effective interaction potential energy includes the nuclear local interaction potential energy and Coulomb interaction potential energy,

$$U = U_{loc} + U_{Coul} \tag{9}$$

$U_{loc}$ is obtained from the integration of the nuclear local interaction potential energy density functional. The nuclear local interaction potential energy density functional $V_{loc}(\rho(r))$ is taken the same as that in Ref. [26], which reads

$$V_{loc} = \frac{\alpha}{2}\frac{\rho^2}{\rho_0} + \frac{\beta}{\gamma+1}\frac{\rho^{\gamma+1}}{\rho_0^{\gamma}} + \frac{g_0}{2\rho_0}(\nabla\rho)^2 + g_t\frac{\rho^{\eta+1}}{\rho_0^{\eta}} + \frac{C_s}{2\rho_0}(\rho^2 - \kappa_s(\nabla\rho)^2)\delta^2 \quad (10)$$

Here $\rho$, $\rho_n$ and $\rho_p$ are the nucleon, neutron, and proton density distributions of system, respectively, and $\delta = (\rho_n - \rho_p)/(\rho_n + \rho_p)$ is the isospin asymmetry. By integrating $V_{loc}$, we obtain the local interaction potential energy

$$U_{loc} = \frac{\alpha}{2}\sum_{i,j\neq i}\frac{\rho_{ij}}{\rho_0} + \frac{\beta}{\gamma+1}\sum_{i,j\neq i}\left(\frac{\rho_{ij}}{\rho_0}\right)^{\gamma} + \frac{g_0}{2}\sum_{i,j\neq i}f_{s(ij)}\frac{\rho_{ij}}{\rho_0} + g_\tau\left(\frac{\rho_{ij}}{\rho_0}\right)^{\eta} + \frac{C_s}{2}\sum_{i,j\neq i}t_{iz}t_{jz}\frac{\rho_{ij}}{\rho_0}\left(1 - \kappa_s f_{s(ij)}\right) \quad (11)$$

$$\rho_{ij} = \frac{1}{(4\pi\sigma_r^2)^{3/2}}\exp\left(-\frac{(\vec{r}_i - \vec{r}_j)^2}{4\sigma_r^2}\right) \quad (12)$$

$$f_{s(ij)} = \frac{3}{2\sigma_r^2} - \left(\frac{r_{ij}}{2\sigma_r^2}\right)^2 \quad (13)$$

and $t_{iz} = 1$ and $-1$ for proton and neutron, respectively.

The Coulomb energy is written as the sum of the direct and the exchange contribution,

$$U_{coul} = \frac{1}{2}\int\rho_P(\vec{r})\frac{e^2}{|\vec{r} - \vec{r}'|}\rho_P(\vec{r}')d\vec{r}d\vec{r}' - e^2\frac{3}{4}\left(\frac{3}{\pi}\right)^{1/3}\int\rho_P^{4/3}dR \quad (14)$$

The phase space occupation constraint method and the system-size-dependent wave-packet width are adopted as those in the previous version of the ImQMD model [18,19]. The parameters used are the same as in Ref. [26] (see Table 1).

**Table 1.** Model parameters.

| Para. | $\alpha$ MeV | $\beta$ MeV | $\gamma$ | $g_\tau$ MeV | $g_0$ MeVfm$^2$ | $\eta$ | $C_s$ MeV | $\kappa_s$ fm$^2$ | $\rho_0$ fm$^{-3}$ |
|---|---|---|---|---|---|---|---|---|---|
| IQ2 | −356 | 303 | 7/6 | 12.5 | 7 | 2/3 | 32 | 0.08 | 0.165 |

The proper initial condition, which makes the initial nuclei in the real ground state, is of importance because considerable excitation of initial nuclei will cause spurious nucleon emission and affects the products of low-energy nuclear reactions. Before studying the reactions for very heavy nuclei using the ImQMD model, a large number of tests for the model from many aspects is required. In Figure 1, we present the time evolution of binding energies and root-mean-square charge radii for $^{144}$Nd, $^{156}$Dy, $^{197}$Au, $^{238}$U, and $^{250}$Cf calculated by the ImQMD model with a parameter set of IQ2. One can see that their binding energies and root-mean-square charge radii remain constants with a very small fluctuation and the bound nuclei evolve stably without spurious emission for a period of time of about 6000 fm/c, which is essential for applications to fusion and quasi-fission reactions of heavy nuclei. For example, we elaborately select 20 projectiles and targets initial $^{197}$Au nuclei from thousands of sampled $^{197}$Au nuclei by using the ImQMD model. The binding energy for $^{197}$Au is required to be $7.92 \pm 0.05$ MeV/nucleon, and its root-mean-square radius is required to be $5.44 \pm 0.2$ fm, and the bound nuclei evolve stably without spurious emission. More than 100,000 collisions are simulated in all, by rotating these prepared projectile and target nuclei around their centers of mass by an Euler angle chosen randomly. The distance from the projectile to the target at an initial time is taken to be 50 fm.

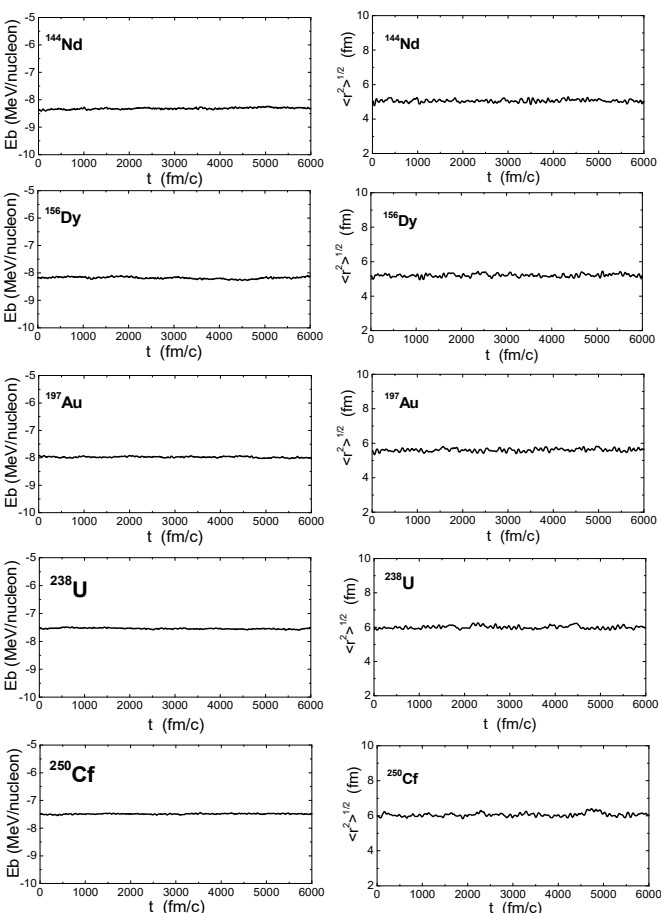

**Figure 1.** The time evolution of binding energies and root-mean-square charge radii for $^{144}$Nd, $^{156}$Dy, $^{197}$Au, $^{238}$U and $^{250}$Cf calculated by the ImQMD model.

## 3. Ternary and Quaternary Breakup in Collisions of Two Massive Nuclei

As in experiment [1–4], a class of ternary events satisfying nearly complete balance of mass numbers is selected under the condition allowing for emit nucleons up to 70 mass units, i.e.,

$$A_p + A_T - 70 \leq A_1 + A_2 + A_3 \leq A_P + A_T \tag{15}$$

where $A_P + A_T$ is the total mass number and $A_1$, $A_2$, and $A_3$ are the masses of three fragments, respectively. Further, the conditions on the balance of longitudinal and transversal momentum applied in the experiment of Refs. [2–4] to make a further selection for the events in the calculations is also adopted in the event selection, $\left| \sum_{i=1}^{3} \vec{P}_{\text{long}}(i) \right| > 0.8 P_0$ and $\left| \sum_{i=1}^{3} \vec{P}_{trans}(i) \right| < 0.04 P_0$, where $P_0$ is the momentum of $^{197}$Au projectiles. By counting the number of $A_1$, $A_2$, and $A_3$ masses at each impact parameter $b$, the production cross sections for $A_1$, $A_2$, and $A_3$ are obtained with the expression

$$\sigma(A_i) = 2\pi \int_0^{b_{\max}} b P(A_i, b) db, \tag{16}$$

where $P(A_i, b) = N(A_i, b)/N_0(b)$, ($i$ = 1, 2, 3) is the production probability of fragment $A_i$ with impact parameter $b$. $N(A_i, b)$ and $N_0(b)$ denote the number of fragments $A_i$ produced in ternary events and the total ternary breakup events with impact parameter b. Here, $b_{\max}$ and $b$ are 12.0 and 1.0 fm, respectively.

### 3.1. The Mass Distributions and Angular Distributions of Ternary Breakup Fragments

The comparison between the calculation results and experimental data for the mass number distributions is shown in Figure 2 for three fragments $A_1$ (the heaviest fragment), $A_2$ (the intermediate fragment), and $A_3$ (the lightest fragment) in the ternary breakup reactions of $^{197}$Au+$^{197}$Au at the energy of 15 $A$ MeV. As we can see from the figure, the most probable ternary events involve the formation of three comparable fragments. The peak of mass distribution for the third fragment $A_3$ was found to locate at about 100. Such a large mass difference seems difficult to explain by the normal nucleon transfer mechanism.

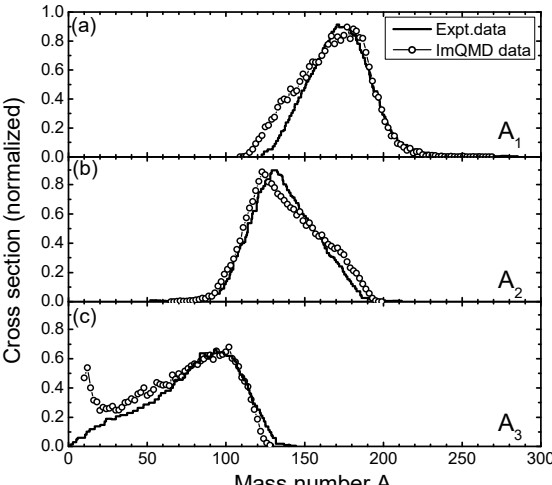

**Figure 2.** Mass number distributions of (**a**) the heaviest $A_1$, (**b**) middle-mass $A_2$, and (**c**) the lightest $A_3$ fragments in selected ternary reactions of $^{197}$Au+$^{197}$Au at an energy of 15 A MeV. The experimental data taken from Ref. [1] with the histogram.

In the reaction process, a transient composite system may be formed due to the strong dissipation, and then the formed transient composite system elongates and breaks up into two parts namely, the projectile-like fragment (PLF) and the target-like fragment (TLF) followed by a further breakup of PLF (or TLF) after a short time, leading to a ternary breakup reaction. A similar mechanism can be extended to quaternary reactions. (see Section 3.6). The reaction plane is defined by the beam direction and the separation axis of the PLF and the TLF, which is the direction of the vector of $\vec{V}_{PT} = \vec{V}_{PLF} - \vec{V}_{TLF}$ with $\vec{V}_{PLF}$ and $\vec{V}_{TLF}$ being the velocities of the projectile and target in the laboratory system, respectively. The definition is the same as in Ref. [4]. Figure 3 shows the results of the out-of-plane angle $\theta$, the azimuthal angle $\varphi$, and as well as the angle $\theta_{\text{c.m.}}$ (between the beam direction and $\vec{V}_{PT}$) distributions of fragments from PLF $\to$ F$_1$ + F$_2$ breakup obtained from the ImQMD model simulations (the lines with solid circles). The experimental results from Ref. [4] are also shown as red lines with triangles. One can see from the figure that the most of ternary breakup events are in the reaction plane and three fragments are approximately aligned. Figures 2 and 3 clearly show us the calculation results can reproduce the experimental results nicely, and it tells us that the ImQMD model provides us with a desirable approach to the study of the mechanism of ternary breakup reactions. However, how are these three fragments produced and which ingredients affect their production probability?

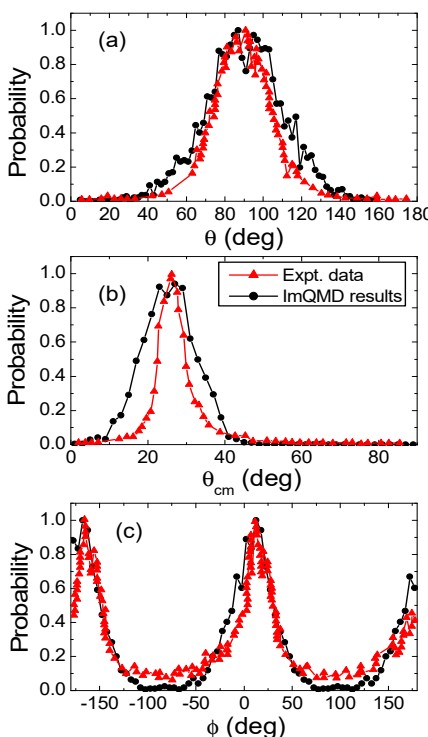

**Figure 3.** Angular distributions of fragments (**a**) out-of-plane angle $\theta$, (**b**) $\theta_{\text{c.m.}}$, and (**c**) azimuthal angle $\varphi$ in cascade ternary reactions. The experimental data take from Ref. [4] with red triangles.

*3.2. Production Probability of Ternary Reactions*

Firstly, the impact parameter plays a crucial role in this process, it not only affects the ternary mode but also decides the production probability of ternary events. Figure 4 presents the entrance channel dependence of production probability of ternary events for forming the same composite system consisting of projectile and target with two different reactions $^{197}$Au+$^{197}$Au (solid square) and $^{156}$Dy+$^{238}$U (solid circle) at the same center-of-mass energy 1478 MeV. One sees that the behavior of production probability of ternary events of the composite systems for both reactions is quite similar, and depends on the impact parameters. The production probability is increasing with the impact parameter increase from 0 to 3 fm, and it reaches the highest and keep a stability value approaching 0.3 in the region $b$ = 3–7 fm, then a rapid descent from 7 to 12 fm for $^{197}$Au+$^{197}$Au. For reaction system $^{156}$Dy+$^{238}$U, the production probability is increasing with the impact parameter increase from 1 to 5 fm, and it reaches the highest value of 0.27 at $b$ = 5 fm, then a rapid descent from 6 to 12 fm. The probability of producing ternary events is smaller for asymmetric reaction system $^{156}$Dy+$^{238}$U than that in symmetric reaction system $^{197}$Au+$^{197}$Au. Maybe the reaction Q value plays an important role, the reaction Q value is –653 MeV for $^{197}$Au+$^{197}$Au, while the reaction Q value is –614 MeV for $^{156}$Dy+$^{238}$U. So, the translated relative kinetic energy is larger than that for $^{156}$Dy+$^{238}$U. It implies that the semi-central collisions are beneficial for producing ternary events for both reaction systems.

The second ingredient is incident energy which affects the production probability of ternary events. Figure 5 shows the energy dependence of the production probability of ternary events for $^{197}$Au+$^{197}$Au (solid square) and $^{156}$Dy+$^{238}$U (solid circle) systems with impact parameter $b$ = 1 fm. It shows that the production probability of ternary events is increasing with energy increase from 5–24 $A$ MeV for $^{197}$Au+$^{197}$Au, and reaches the highest value approaching 0.6 at about 24 $A$ MeV, then a rapid descent from 24 to 30 $A$ MeV due to increasing of the quaternary breakup events. This implies very important information. At this certain energy region, ternary fission in addition to binary fission becomes dominant due to the high excitation energy, which provides a reference value for the experimental scientists in this aspect. In a certain sense, the ternary fission phenomenon in heavy ion

reactions for heavy systems is a phenomenon between multifragmentation and fusion fission. At $E = 5\,A$ MeV, ternary events are not found because the incident energy is below the Coulomb barrier [27] (662 MeV) of $^{197}$Au+$^{197}$Au and no composite system formation. Though binary fission is dominant, the ternary events are increasing with the increase of the incident energy from 10 to 20 A MeV. At the energy region, $E = 24–30\,A$ MeV the probability of ternary breakup is decreasing since multifragmentation events (more than 3 fragments) become more and more. It indicates that the production probability of ternary breakup is very sensitive to the incident energy, and the incident energy near 24 A MeV is most beneficial for producing ternary events for $^{197}$Au+$^{197}$Au with $b = 1$ fm. The same case can be seen for $^{156}$Dy+$^{238}$U.

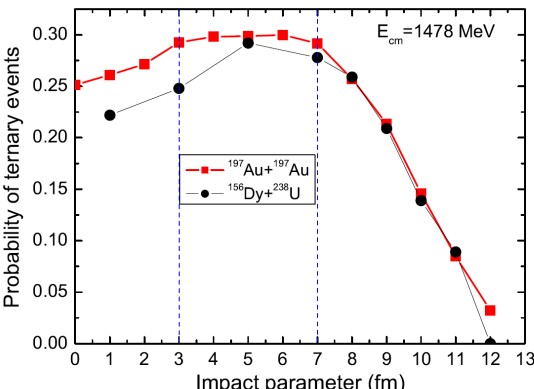

**Figure 4.** Entrance channel dependence of production probability of ternary events for same composite system consist of projectile and target with different reaction $^{197}$Au+$^{197}$Au (solid square) and $^{156}$Dy+$^{238}$U (solid circle) at the same center-of-mass energy 1478 MeV.

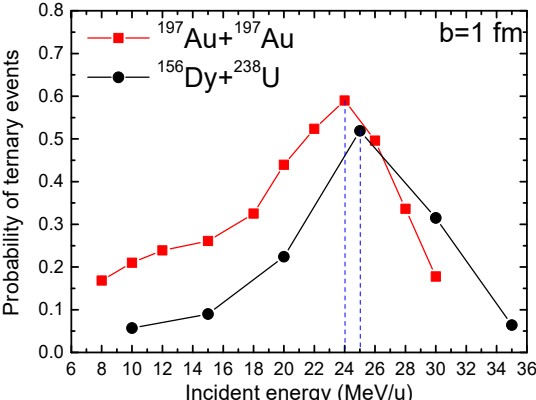

**Figure 5.** The incident energy dependence of the production probability of ternary events for the same composite system of different reaction $^{197}$Au+$^{197}$Au and $^{156}$Dy+$^{238}$U with $b = 1$ fm.

### 3.3. The Ratio of N/Z for the Third Fragment $A_3$

As seen from the above study, the N/Z ratio of the third fragment $A_3$ is one of the most sensitive quantities with respect to the neck formation and the origin of the third fragment, as shown in Figure 6. For the isospin symmetry case of Au+Au, the N/Z ratio is 1.49, and the average value is almost unchanged keep the value 1.49 in the reaction process until the composite system re-separates.

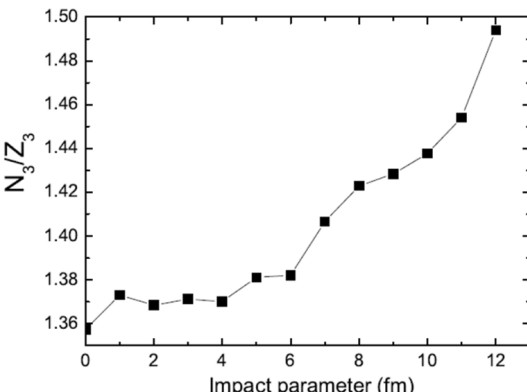

**Figure 6.** The impact parameter dependence of the ratio of neutrons number versus protons number for the third fragments $A_3$.

In Figure 6, we show the impact parameter dependence of the N/Z ratio for the third fragment $A_3$. At the region $b$ = 0–6 fm the N/Z ratio is almost unchanging maintaining a value of about 1.37. The impact parameter strongly influences the N/Z ratio of the third fragment $A_3$, in particular for impact parameter b > 6 fm, in which the ternary mode is mainly a participant-spectator scenario and the third fragment becomes smaller with increasing impact parameters. When $b$ = 12 fm the ratio is reaching to the value of 1.49, the same as the ratio of Au, whereas the average mass number of $A_3$ is about 20, which is relative to the ratio in the stable nuclei with the same mass in β-stability line is very neutron-richer. This may be an approach to produce the very neutron-rich isotope in the experiment. This effect results from the different behavior of the density dependence of the chemical potential for neutrons and protons in reaction systems [28]. In which the time starts from the beginning of the neck formation when the density at the touching point reaches $0.02\rho_0$. The reason may be understood as follows: at the beginning when the neck is just formed, neutrons preferably move to the neck region driven by the chemical potential, not soon, as too many neutrons are concentrated there, the symmetry potential attracts more protons to migrate into the neck region and the N/Z ratio is reduced; then, because of the increase of the proton number the Coulomb repulsion plays a role. Thus the interplay of the Coulomb force and the symmetry potential results in a fluctuation behavior in the N/Z ratio for participants from the neutron-rich third fragment at the neck region.

*3.4. The Space Distributions of Three Fragments in Ternary Breakup Reactions*

Now we turn to study the spatial distributions of three fragments in the ternary breakup of the giant composite system transiently formed by reaction $^{197}$Au+$^{197}$Au. There are four possible modes by which three comparable mass fragments can be produced in heavy-ion-induced fission. These are sketched in Figure 7. It has already been shown by Diehl and Greiner [29] that there are two possible direct modes, the oblate and prolate ones for fission into three fragments, and cascade fission mode. In a direct prolate ternary event, two necks are preformed almost simultaneously, and their centers are almost in alignment. In the direct oblate ternary event, when the necks are formed the initial configuration of three fragments centers of mass is near an equilateral triangle. In cascade (or sequential) fission, a heavy fragment produced in normal binary fission may have sufficient excitation energy to also fission subsequently. In our microscopically dynamical study, time-dependent density contour plots can allow us to identify the different fission mechanisms (fission modes) and extract the corresponding fission time scales. The time interval $t_{2\text{-}1}$ between the first and second separation in ternary events is defined for understanding the mechanism of ternary breakup reactions. For the direct ternary process, the time interval $t_{2-1}$ is much smaller than 100 fm/c and we cannot show it here because the time interval for recording the position and momenta of particles is 100 fm/c in the

calculations, given that the two separations happen almost simultaneously. In Ref. [13] only prolate and cascade ternary breakup events were studied.

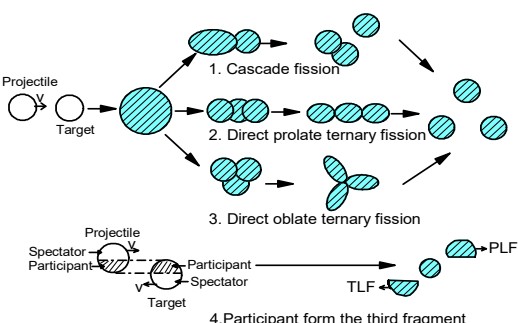

**Figure 7.** Possible modes of ternary breakup reactions.

Figure 8 shows a snapshot of a typical direct prolate ternary breakup event at different times for the case of central collisions of $^{197}$Au+$^{197}$Au. From this figure, one sees that after touching time ($t$ = 300 fm/c, see Figure 8b), two participants deeply interact and some compression may take place with the system heating up, then an expansion follows and an elongation takes place along the axis roughly perpendicular to the beam direction; about $t$ = 1600 fm/c, two necks (Figure 8h) seem to be formed at almost the same time, then the two necks break up sequentially in a very short time interval. The time scale from the formation of the composite system to fission into three fragments is about 1500 fm/c. However, the time interval from the first partition to the second one is very short, about $t_{2\text{-}1}$ = 100 fm/c. The experimental characteristic features in ternary breakup events, three mass-comparable fragments, and the very fast, nearly collinear breakup, maybe stem from this special type configuration of the composite system which has preformed two necks. Though for the cascade ternary breakup event, the time interval is very long, about $t_{2\text{-}1}$ = 1300 fm/c, the process of time evolution is shown in Figure 9. For only one neck preformed case the residue after first fission takes a long time to rearrange the particle to reach a lower energy state and the system continues to elongate and finally separate. For the prolate ternary breakup events produced by semi-central collisions (for example, impact parameter $b$ = 6 fm), the time evolution of density distribution for a ternary event is different from the case in the central collision due to an amount of angular momentum. In this case, the elongation axis of the composite system rotates with respect to the beam direction. The formation and rupture of two necks in the composite system are also almost simultaneous. The produced three fragments are along the elongation axis. The time scale for this type of ternary breakup event is shorter with increasing impact parameters.

Figure 10 shows the time evolution of a typical oblate ternary breakup event produced at central collisions. This fission mode is very interesting although its production probability is very low. This mode may have special importance from the point of view of nuclear structure study due to the exotic configuration family. At t~300 fm/c, the interacting nuclei begin to stick together and form a compact mononuclear system (see Figure 10c–f) and keep this shape at about 1200 fm/c. Then the system expands and deforms to a triangle-like configuration (see Figure 10g). Figure 10h shows a very exotic configuration with a symmetric three preformed necked-in. At about $t$ = 1800 fm/c (see Figure 10i), the three necks rupture almost simultaneously and the composite system breaks up into three equally sized fragments along space-symmetric directions in the reaction plane. The time scale for this kind of oblate ternary breakup is larger than that for the prolate ternary breakup process.

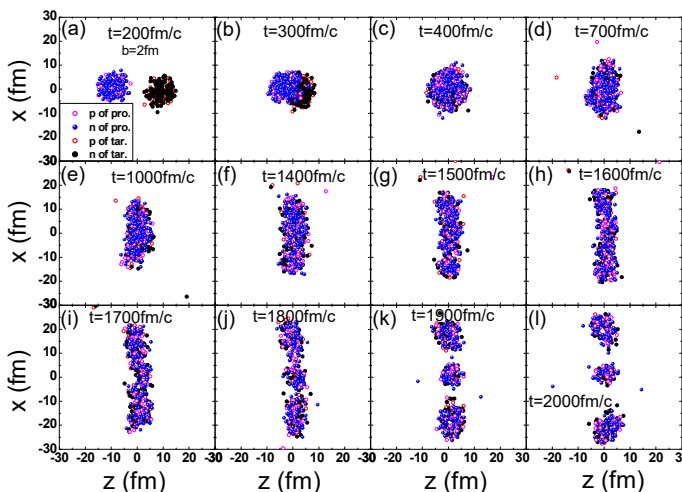

**Figure 8.** The time evolution of a direct prolate ternary event for $^{197}$Au+$^{197}$Au at 15 $A$ MeV with $b = 2$ fm.

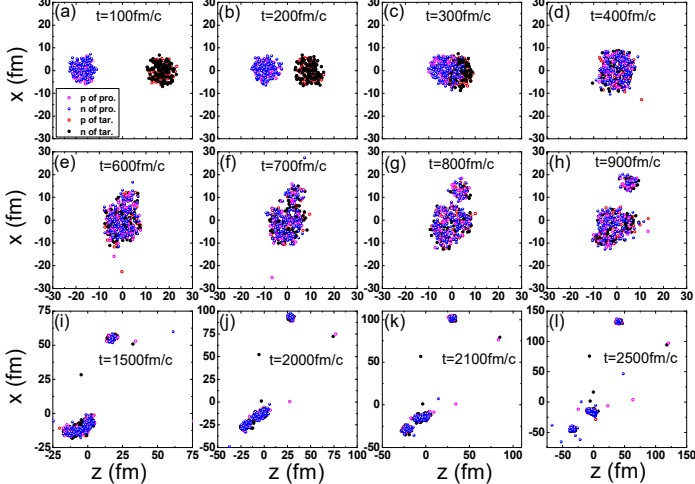

**Figure 9.** The time evolution of a cascade ternary event for $^{197}$Au+$^{197}$Au at 15 $A$ MeV with $b = 2$ fm.

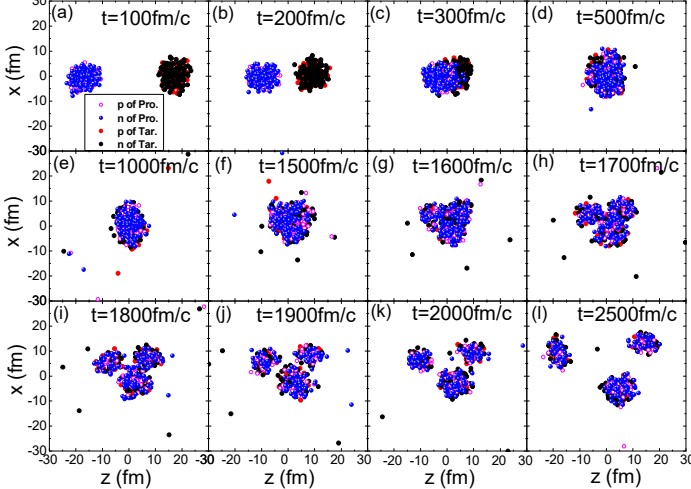

**Figure 10.** The time evolution of a direct oblate ternary event for $^{197}$Au+$^{197}$Au at 15 $A$ MeV with $b = 2$ fm.

Figure 11 shows the impact parameter dependence of the production probabilities for cascade, prolate and oblate ternary breakup events in the reaction $^{197}$Au+$^{197}$Au at 15 $A$ MeV. The probability of the cascade mode is the largest in ternary breakup events at central and semi-peripheral reactions. The probability of prolate ternary breakup events increases, and that for cascade ternary reactions decreases with increasing the impact parameters, and at very large impact parameters, the probability of prolate ternary events exceeds that for cascade ternary events. It is clear that at very large impact parameters (peripheral reactions), most of the third fragments come from the neck and the mass of the third fragment decreases with increasing the impact parameter. We will see that the mechanism of ternary breakup changes from central to peripheral collisions.

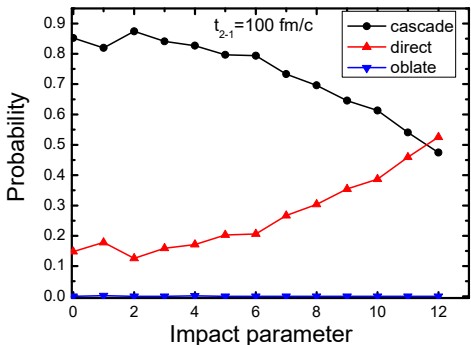

**Figure 11.** The production probabilities depend on the impact parameter for cascade, prolate and oblate ternary breakup events for $^{197}$Au+$^{197}$Au at 15 $A$ MeV.

### 3.5. Probing the Energy Dissipation Mechanism of Ternary Breakup Reactions

The features observed in the ternary breakup reaction between two $^{197}$Au nuclei indicate that strong dissipation plays an important role in the reaction process, and the deep study of the ternary breakup can help us understand the interplay between the one-body or two-body dissipation mechanism. In order to clarify the mechanism of nuclear energy dissipation, we first calculate the time evolution of the translation kinetic energy in the relative motion for the reaction $^{197}$Au+$^{197}$Au. We find that before the touching of the projectile and target, a part of the translation kinetic energy converts into potential energy and excitation energy of the system. At about 600 fm/c, almost all translation kinetic energy in relative motion dissipates, and the excitation energy reaches the largest value. According to the definition of the two-body dissipation function in hydrodynamics, i.e., $\frac{-dE}{dt} = \sum_{i,j} \gamma_{i,j} \dot{q}_i \dot{q}_j$, we can roughly estimate the two-body viscosity for relative motion. Figure 12 shows the translation kinetic energy loss as a function of the square of the velocity of the relative motion. When only a relative motion is considered, the two-body viscosity can be estimated. It is about $10^{-21}$ MeV s fm$^{-2}$, which is a quite strong dissipation. The large kinetic energy loss leads to the high internal excitation of the colliding system, and the composite system subsequently happens prompt decay.

On the other hand, we choose the mean-free path of nucleons to probe the energy dissipation mechanism. One-body nuclear dissipation connects with the long mean-free path of nucleons inside a nucleus, which arises from nucleons colliding with the moving potential wall rather than with another nucleon [5,6]. Two-body dissipation proceeds from collisions between individual nucleons, which should apply only to systems for which the mean free path is smaller compared to the spatial dimensions [7,8]. The mean free path of nucleons is calculated in the period from touching configuration to the composite system re-separation for each event. The nucleon-nucleon collision times are memorized in the ImQMD model. The mean-free path is defined by $\lambda = \frac{1}{A} \sum_{i=1}^{A} \lambda(i)$, where $\lambda(i)$ is the mean-free-path of the $i$th nucleon and $A = A_{\mathrm{P}} + A_{\mathrm{T}}$ is the total nucleons. In the calculation, we trace the path of each nucleon in the reaction system from the formation

of composite systems to their re-separation and measure the length of the path between every two sequential collisions. By summing up the total lengths of paths and counting the number of collisions, one can calculate λ(i) for each nucleon and obtain the mean-free path. The correlation between the mean-free path and the mass number is investigated in the third fragment $A_3$. In the simulation of the ImQMD model, the mass number of fragment $A_3$ and the mean-free-path of nucleons can be obtained simultaneously for each ternary event. Then, the statistical average value of the mean-free path of nucleons is calculated for those events producing the same mass fragment A₃. Figure 13 shows the mean-free path of nucleons in a composite system as a function of the third fragment mass $A_3$. From Figure 13, we can see that the mean-free path decreases with the increasing $A_3$ mass until the region the average mass of $A_3$ = 85–105, where it becomes flatter. The figure clearly shows that the mean-free-path in the ternary fission process to produce the large mass $A_3$ is much shorter than that producing the smaller mass $A_3$. Thus, the mean-free path shortens with the increasing number of nucleon-nucleon collisions leading to the ternary process happening to produce three comparable mass fragments. In this case, the two-body energy dissipation mechanism will play a significant role. With the decrease of the mass number of the third fragment $A_3$, the mean-free path increases considerably and becomes comparable with the system size. In this case, the effect of the one-body dissipation mechanism becomes dominant. We conclude from the correlation between the mean-free path and the mass number of A₃ that the role of one-body dissipation becomes weaker and two-body dissipation will be dominant with the increase of mass number A₃. This microscopic calculation seems to support the conclusion of Carjan's in Ref. [9], the mass distribution of the third fragment $A_3$, displayed at the bottom of Figure 2, can be interpreted as an indication of the dominance of two-body dissipation mechanism in the observed ternary fission of the Au+Au system. This result of the third fragment with a relatively large mass is considered experimental evidence for clarifying the competition between one-body and two-body dissipation processes and understanding the microscopic dynamics of those two dissipation mechanisms.

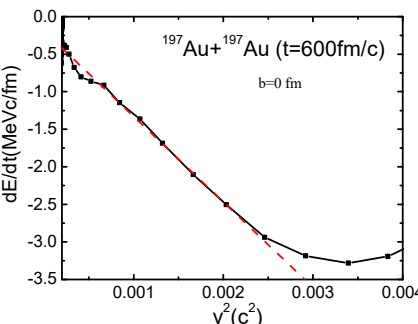

**Figure 12.** Translation kinetic energy loss of relatively collective motion as a function of square of velocity of relatively collective motion for the system $^{197}$Au+$^{197}$Au at 15 *A* MeV. The red dashed line is a guide to the eye.

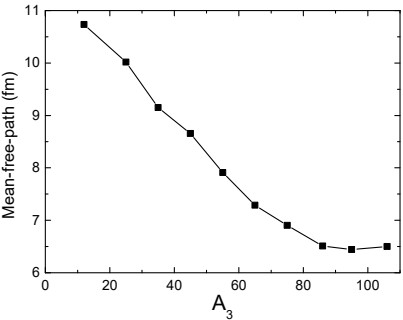

**Figure 13.** Mean free path of average each nucleon in composite systems is shown as a function of the third fragments mass $A_3$.

### 3.6. The Modes and Mechanisms of Quaternary Breakup Reactions

Quaternary partitioning of heavy colliding systems has also been reported in the past, but the experimental phenomenon of the dynamical quaternary breakup is first demonstrated in Ref. [2] by I. Skwira-Chalot et al. A new mechanism was found for the reaction system of $^{197}$Au+$^{197}$Au quaternary breakup into four aligned fragments of comparable size. The difficulty in the experiment is that one has to distinguish between simultaneous and sequential quaternary decay. They assumed a binary process in the primary stage: $^{197}$Au+$^{197}$Au → TLF + PLF, followed by secondary decay processes: PLF → $F_1$ + $F_2$ and TLF → $F_3$ + $F_4$. Although what is known about the quaternary reaction mechanism? A brief presentation of the quaternary reactions is essential because these reactions represent a natural extension of the mechanism of ternary partitions. Although quaternary events are very rare, only 56 quaternary breakup events are found in 100,000 ImQMD simulation events. We have also found three types of quaternary breakup modes (direct prolate, oblate, and cascade quaternary) in the simulation processes.

The quaternary breakup process can appear in two different ways simultaneous and sequential quaternary breakup. (i) the simultaneous creation of four fragments in the act of fission (see Figure 14, it convincingly demonstrates nearly collinear partition of all four fragments), and (ii) via a fast sequential decay of one of three fragments with high excitation energy after the ternary breakup (see Figure 15).

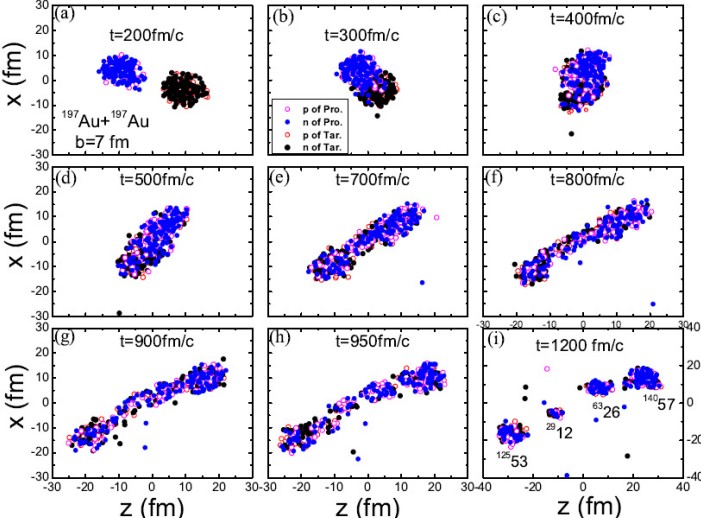

**Figure 14.** The time evolution of a typical direct prolate quaternary breakup event for the $^{197}$Au+$^{197}$Au at energy of 15 *A* MeV and *b* = 7 fm. The open and solid circles represent protons and neutrons, respectively.

Figure 16 shows the time evolution of a typical oblate quaternary breakup event produced at central collisions of U+U. This quaternary mode is very similar to the oblate ternary breakup mode (see Figure 10). Three necks form and rupture almost simultaneously and the composite system breaks up into four equally sized fragments along space-symmetric directions in the reaction plane. This oblate quaternary event is even rarer, which is not found in the reaction Au+Au system but an event is found in 10,000 the simulation $^{238}$U+$^{238}$U reaction events at the energy of 15 *A* MeV and *b* = 0 fm. Another rare process is a pseudo quaternary breakup in $^{197}$Au+$^{197}$Au at 15 *A* MeV with *b* = 7 fm. In this process, both the PLF and the TLF undergo a similar process of a fast breakup, but two-fragment among them together with one and final become a ternary breakup process. Figure 17 shows an example of a special kind of ternary breakup event, in which via the quaternary breakup process, the middle two fragments merge into a larger one. This case is a special ternary breakup mechanism not mentioned previously.

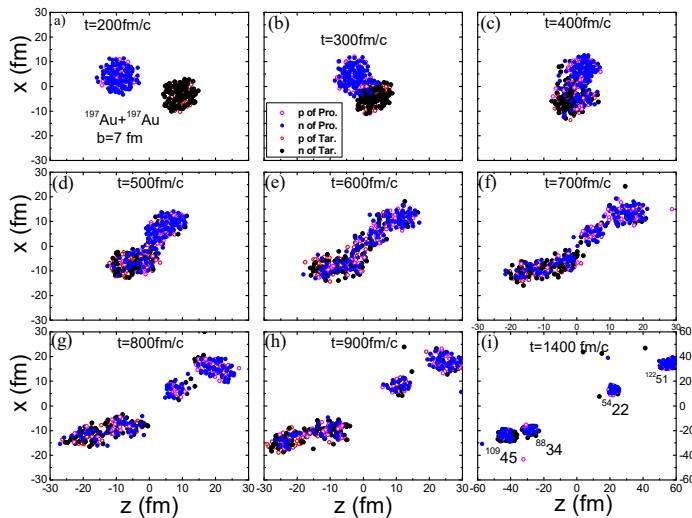

**Figure 15.** The same as Figure 14, but for a typical cascade quaternary breakup event.

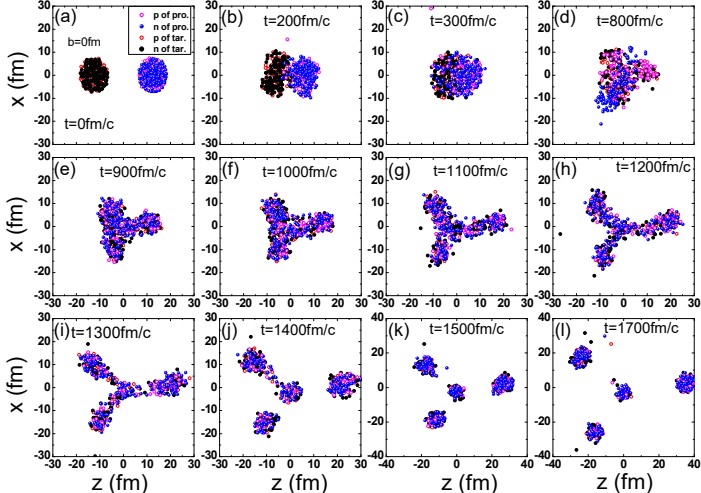

**Figure 16.** The time evolution of a direct oblate quaternary breakup event for $^{238}$U+$^{238}$U at energy of 15 $A$ MeV and $b = 0$ fm.

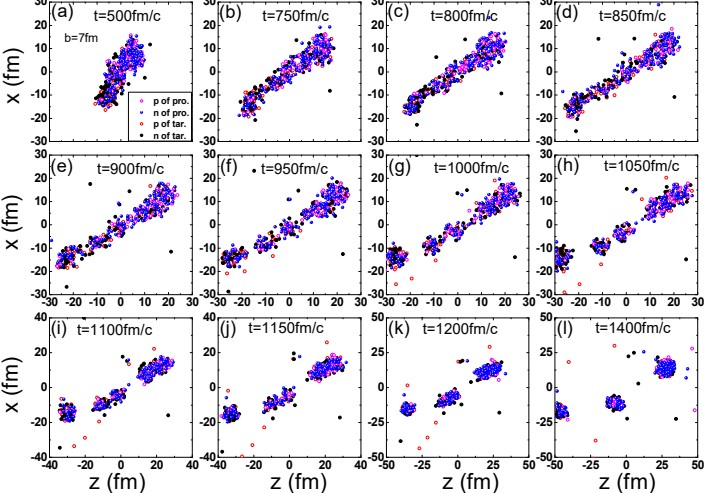

**Figure 17.** The time evolution of a pseudo quaternary breakup event for the middle two fragments merge into a large one in $^{197}$Au+$^{197}$Au reaction at energy of 15 $A$ MeV $b = 7$ fm.

## 4. Conclusions

The mechanism of the ternary and quaternary reactions of the very heavy system $^{197}$Au+$^{197}$Au at the energy range of 5–30 *A* MeV has been studied by using the ImQMD model. The calculation results reproduce the characteristic features in ternary breakup events explored in a series of experiments; i.e., the masses of three fragments are comparable in size and the very fast, nearly collinear breakup. The study shows that the direct prolate ternary mode is responsible for those events having the characteristic features found in the experiments that happen at relatively small impact parameter reactions, not at peripheral reactions, and the configuration of the composite system has a two-preformed-neck shape. The ternary breakup reaction at peripheral reactions belongs to the mode of the participant from the third fragment or binary breakup with a neck emission. It is also found that the probability of ternary breakup depends on the incident energy and the impact parameter. We obtain that the largest probability of ternary breakup is located on the energy around 24 A MeV for the system $^{197}$Au+$^{197}$Au. The modes and mechanisms of ternary and quaternary breakup are studied by time-dependent snapshots of the corresponding events. Three different ternary breakup modes, direct prolate, direct oblate, and cascade are clearly manifested and their production probabilities are given. In a direct prolate ternary event, two necks are preformed and ruptured almost simultaneously, and the three fragment centers are almost aligned. The direct oblate ternary breakup is a very rare event, in which three necks are formed and rupture simultaneously, forming equally sized three fragments along space-symmetric directions in the reaction plane. The direct ternary breakup is an almost simultaneous process, while the cascade ternary breakup is a two-step fission process. In the first step, the reaction system separates into projectile-like and target-like fragments, and in the second step, PLF or TLF breaks into two fragments and the complementary primary fragment survives. For this case, only one neck is preformed of the composite system, the residue after first fission takes a long time to re-arrange the particle to reach a lower energy state and the system continues to elongate and finally separate. For the large parameters, the ternary breakup reaction is dominated by the binary breakup with simultaneously emitted light-charged particles at the neck, and the third fragment mass decrease with increasing impact parameter. In order to clarify the energy dissipation mechanism, the mean-free-path of nucleons in the reaction system is studied and the shorter mean-free-path is responsible for the ternary breakup with three mass-comparable fragments, in which the two-body dissipation mechanism plays a dominant role.

**Author Contributions:** Investigation, J.T. and C.L.; Methodology, J.T. and C.L.; Calculation, X.L.; Writing—Original Draft, J.T.; Writing—Review and Editing, C.L. All authors have read and agreed to the published version of the manuscript.

**Funding:** This work is supported by the National Natural Science Foundation of China (No. 11961131010) and the Central Government Guides Local Scientific and Technological Development Fund Projects (No. Guike ZY22096024).

**Data Availability Statement:** Not applicable.

**Acknowledgments:** The author is grateful to Z.X. Li, X.Z. Wu and S.W. Yan for fruitful discussions.

**Conflicts of Interest:** The authors declare no conflict of interest.

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
