# Peer review of "Improved Quantum Molecular Dynamics Model and Its Application to Ternary Breakup Reactions"

_universe, doi:10.3390/universe8110555_

Round 1

Reviewer 1 Report

The authors Tian, X. Lie and C Li must explain the theoertical basis and equations for the 17 figures.

Author Response

There are 16 equations of throughout the paper, in which Eq. (1) –(14) are the theoretical basis of the ImQMD model, Eq. (15) is the selected condition for ternary events satisfying nearly complete balance of mass numbers, and Eq. (16) is the theoretical basis of the mass number distributions of Fig.2.  17 figures are obtained based on these 16 equations.

Reviewer 2 Report

Referee report on the manuscript # universe-1852135 

by Junlong Tian, Xian Li and Cheng Li 

entitled

"Improved Quantum Molecular Dynamics Model and Its Application to Ternary Breakup Reactions"

In this paper the authors discuss the application of the improved quantum molecular dynamics (ImQMD) model for the description of gold-gold collisions at the energy range of 5 - 30 AMeV. The improvement they made concerns the ternary breakup reactions, which can be subdivided further to the direct prolate, direct oblate, and cascade ternary breakup reactions. Results of the model calculations are confronted to the available experimental data and show a fair agreement. Further investigation is dealing with the mechanisms and modes of ternary and quaternary breakup reactions studied by time-dependent snapshots of ternary events. The mean free path of nucleons in the reaction systems is studied as well.

The paper is clearly written. The model ImQMD is presented in detail, all model parameters are described, and all conclusions are supported convincingly by the figures.

However, I have several questions and remarks.

1. Does this model (ImQMD) contain the liquid-gas phase transition in nuclear matter which should take part at slightly higher collision energies?

2. In Figure 1 we see that the binding energies are only slightly fluctuating over time, while the fluctuations of the root-mean-square charge radii are more pronounced. Why?

3. Figure 2 shows that the mass number distributions of the heaviest (A1) and the middle-mass (A2) fragments are nicely reproduced by ImQMD. As to the lightest-mass fragments (A3), one sees a clear deviation of model calculations from the data in the range of low masses (A < 30). What is the origin of this discrepancy and how it can be improved in future?

4. Caption to Figure 12 misses the explanation of red dashed line in the plot.

5. I suggest to use the similar notations for the mass fragments, i.e. $A_1$, $A_2$ and $A_3$, throughout the paper, instead of a mixture of, e.g., $A_3$ and A3 (see page 13).

After clarification of these issues, I recommend publication of the manuscript in Universe.

Author Response

  1. Does this model (ImQMD) contain the liquid-gas phase transition in nuclear matter which should take part at slightly higher collision energies?

Our reply:

Yes, the ImQMD model can be applied to high energies up to 1GeV/u, these works can be seen in Ref. [28] Ying-Xun Zhang, Ning Wang, Qing-Feng Li, Li Ou, Jun-Long Tian, Min Liu, Kai Zhao, Xi-Zhen Wu, Zhu-Xia Li; Progress of quantum molecular dynamics model and its applications in heavy ion collisions, Frontiers of Physics, 15, (2020) 54301

2.In Figure 1 we see that the binding energies are only slightly fluctuating over time, while the fluctuations of the root-mean-square charge radii are more pronounced. Why?

Our reply:

With time evolution, there is exist a breather mode of the initial nuclei created by the ImQMD model. (see the attached animation file, Au initial nuclei evolution.gif).

In the dynamical process, the root-mean-square charge radii are changing with time evolution within a reasonable range (Delt_r<0.02fm), which leading to the changing of the binding energies. The binding energy is calculated by the integration of the density distributions of nucleon number of the system. The change of the root-mean-square charge radii is not sensitive to the density distributions of nucleon number of the system. So the fluctuations of the root-mean-square charge radii are more pronounced than that of the binding energy.

3.Figure 2 shows that the mass number distributions of the heaviest (A1) and the middle-mass (A2) fragments are nicely reproduced by ImQMD. As to the lightest-mass fragments (A3), one sees a clear deviation of model calculations from the data in the range of low masses (A < 30). What is the origin of this discrepancy and how it can be improved in future?

Our reply:

  For the lightest-mass fragments, the ImQMD model can not describe well due to absent the shell structure effect of the nuclei. The origin of this discrepancy is that the total N-body wave function is assumed to be the direct product of coherent states not anti-symmetrization wave function of the system in the ImQMD model. In order to overcome the difficulty in describing the Fermionic nature of an N-body system in the QMD model, an approximate treatment of anti-symmetrization is adopted, namely, the phase space constraint of the CoMD model proposed by Papa et al.( see Ref. [24] M. Papa, T. Maruyama, A. Bonasera, Constrained molecular dynamics approach to fermionic systems. Phys. Rev. C, 64, (2001) 024612), which is applied to the model. However, the distribution of light particles is still not well described by the model. Maybe in the future, this intractable problem can be tackled by completely taking into account the effect of anti-symmetrization wave function.

  1. Caption to Figure 12 misses the explanation of red dashed line in the plot.

Our reply:

 “The red dashed line is a guide to the eye”, which is added to the Caption to Figure 12 in the revised version manuscript. See Page 12 the red font.

  1. I suggest to use the similar notations for the mass fragments, i.e. $A_1$, $A_2$ and $A_3$, throughout the paper, instead of a mixture of, e.g., $A_3$ and A3 (see page 13).

 Our reply:

 The notations for the mass fragments A1, A2 and A3, are all changed to $ A_1$, $A_2$ and $A_3$ throughout the paper.

Thanks again for your valuable comments and suggestions!

Reviewer 3 Report

Report on “Improved Quantum Molecular Dynamics Model and Its Application to Ternary 

Breakup Reactions”

By Junlong Tian, Xian Li and Cheng Li

In this work authors investigate the rare ternary breakup reactions of two heavy ion collisions in 

which the final products are three nuclear fragments with comparable masses.

This is very interesting subject and authors succeeded to describe many characteristics 

of the reaction and in my opinion the work deserves a publication.

However I have few questions which I hope authors will clarify in the revised version 

of the manuscript before it will be accepted for publication.

1. Authors call their approach as improved quantum molecular dynamics. However 

it is not clear compared to what deficiencies of standard molecular dynamics they improved 

their model. What was the problem of QMD that they improved.  

2. Model completely ignores the shell structure of the nuclei, while it is well know that fission 

processes are very sensitive to the shel structure. How they can explain this?  After all it is known 

that excitation of fragments or intermediate states depend on the distribution of nucleons in

particular shells.

3. It will be helpful if authors elaborate the anti-symmetrization procedure as well as how they 

took into account Pauli blocking effects in the reaction.

4. Coming to the analysis of the reaction, they observed the peak at 24 MeV incident energy, however 

it seems to me that no explanation is given for existence of such a peak. Can they predict 

the magnitude of similar peaks for ternary break up in collision of other heavy nuclei?

Such a prediction will be the best test of the validity of the model they are discussing.

5. Finally, it is not clear  what conclusion authors arrived for the  space distribution of 

the ternary break-up (Fig.7) is it scenario 4 or 2 that dominates the break-up process?

It will be helpful if clear statement will be given about dominating mechanism of the break-up.

Author Response

Dear Referee,

The authors would like to acknowledge the Referee’s valuable comments and suggestions on our manuscript.

We thoroughly revised our manuscript taking into account the Referee’s suggestions. We thank the Referee for these important suggestions and criticisms, that helped us to improve our manuscript substantially. In particular, we have improved Fig.5, and added an extra the explanation of red dashed line in Caption to Figure 12. The space distribution of the ternary break-up (Fig.7) scenario 2 is dominating ternary break-up mechanism, which is used to explain the characteristic features in ternary breakup observed in experiment, three comparable fragments in size and the very fast, nearly collinear breakup events. We found that two necks are preformed and ruptured almost simultaneously in the process of ternary breakup in scenario 4 or 2 in Fig.7 by used the ImQMD model. And the statements are added to the revised version manuscript.

All modifications are marked as red for your convenience of the check. We believe that the amended version of our manuscript responses properly on the Referee’s comments, and corresponds to the level required for publication in Universe. We would like you to consider the revised manuscript for possible publication in Universe.

Yours sincerely,

all authors.

REPLY THE REFEREE3’S COMMENTS

  1. Authors call their approach as improved quantum molecular dynamics. However it is not clear compared to what deficiencies of standard molecular dynamics they improved their model. What was the problem of QMD that they improved.

Our reply:

The main difficulty of the standard QMD model apply to low-energy reactions is the stability of the time evolution of the initial nuclei. One needs a model which can describe not only the ground-state properties of individual nuclei at initial time well but also their time evolution without spurious particle emission. The main improvements are introduced and their effects are analyzed in Ref. (Ning Wang, Zhuxia Li, and Xizhen Wu,PHYS. REV. C, 65, 064608 (2002)).

  1. Model completely ignores the shell structure of the nuclei, while it is well know that fission

processes are very sensitive to the shell structure. How they can explain this?  After all it is known that excitation of fragments or intermediate states depend on the distribution of nucleons in particular shells.

Our reply:

Fission processes are very sensitive to the shell structure. But the excitation energy of the composite system is very high for the reaction system 197Au+197Au at 15 A MeV. The excitation energy is about E*=Ecm+Q=825 MeV, which lead to the ternary fission process is not sensitive to the shell structure. But for the distribution of the lightest-mass fragments is still not well described by the model due to absent the shell structure effect of the nuclei.(see Fig.2c)

  1. It will be helpful if authors elaborate the anti-symmetrization procedure as well as how they took into account Pauli blocking effects in the reaction.

Our reply:

In the ImQMD model, each nucleon is represented by a coherent state, and the total N-body wave function is assumed to be the direct product of coherent states. The anti-symmetrization of the wave function of the system is not considered in the ImQMD model.

In order to overcome the difficulty in describing the Fermionic nature of an N-body system in the QMD model, an approximate treatment of anti-symmetrization is adopted, namely, we implement the phase space constraint of the CoMD model proposed by Papa et al.( see Ref. [24] M. Papa, T. Maruyama, A. Bonasera, Constrained molecular dynamics approach to fermionic systems. Phys. Rev. C, 64, (2001) 024612) into the model.

This is required by the constraint that the one body occupation number in a volume h^3 of phase space centered at (ri ,pi), corresponding to the centroid of the wave packet of particle i, should always be not larger than 1 according to the Pauli principle. The one body occupation number is calculated by

where si and τi  are the third components of the spin and isospin of particle i. We have made a check of the time evolution of individual nuclei and found that by using the procedure of phase space constraint, the requirement is reasonably satisfied and the phase space distribution is efficiently prevented from evolving into a classical distribution from the initial nuclear ground state distribution for a long enough time.

  1. Coming to the analysis of the reaction, they observed the peak at 24 MeV incident energy, however it seems to me that no explanation is given for existence of such a peak. Can they predict the magnitude of similar peaks for ternary break up in collision of other heavy nuclei?

Such a prediction will be the best test of the validity of the model they are discussing.

Our reply:

There is still exist the peak at 25 A MeV for reaction system Dy+U (see Fig.1 as follows). From Fig.2 one can see that the binary breakup events is dominant at the energy region E=10-20 A MeV, the probability of binary events is decreasing, while the probability of ternary events is increasing with increasing incident energy. Ternary events become dominant due to quaternary events added and binary events reduced at the energy of 25 A MeV. Quaternary events become dominant due to ternary events and binary events further decrease at the energy higher than 25 A MeV. The emergence of the peak in ternary events is related to the competition the binary and quaternary added to ternary events. we have improved Fig.5  and the statements are added to the revised manuscript. See Page 8 the red font.

Fig.1 The incident energy dependence of the production probability of ternary events for same composite system of different reaction 197Au+197Au and 156Dy+238U system with b=1 fm, respectively,

 Fig.2 The incident energy dependence of the production probability of binary, ternary and quaternary events for 156Dy+238U system with b=1 fm, respectively,

  1. Finally, it is not clear what conclusion authors arrived for the space distribution of the ternary break-up (Fig.7) is it scenario 4 or 2 that dominates the break-up process? It will be helpful if clear statement will be given about dominating mechanism of the break-up.

Our reply:

The space distribution of the ternary break-up (Fig.7) scenario 2 is dominating ternary break-up mechanism, which is used to explain the characteristic features in ternary breakup observed in experiment, three comparable fragments in size and the very fast, nearly collinear breakup events. We found that two necks are preformed and ruptured almost simultaneously in the process of ternary breakup in scenario 4 or 2 in Fig.7 by used the ImQMD model. And the statements are added to the revised version manuscript. See Page 16 the red font “The study shows that the direct prolate ternary mode is responsible for those events having the characteristic features found in the experiments happen at relatively small impact parameter reactions, not at peripheral reactions, and the configuration of the composite system has two-preformed-neck shape. The ternary breakup reaction at peripheral reactions belongs to the mode of participant from the third fragment or binary breakup with a neck emission.”

Round 2

Reviewer 1 Report

The title of the article that I received is ...Ternary Breakup Reactions

The title in the email that I received is ...ternary reaction of Au+Au

I believe that Junlong Tian and co-authors should change the

title before it is published in Universe